# Time-dependent computational model of post-traumatic osteoarthritis to estimate how mechanoinflammatory mechanisms impact cartilage aggrecan content

Atte S. A. Eskelinen[1]*, Joonas P. Kosonen[1], Moustafa Hamada[1], Amir Esrafilian[1], Cristina Florea[1], Alan J. Grodzinsky[2], Petri Tanska[1], Rami K. Korhonen[1]

1 Department of Technical Physics, University of Eastern Finland, Kuopio, Finland, 2 Departments of Biological Engineering, Electrical Engineering and Computer Science, and Mechanical Engineering, Massachusetts Institute of Technology, Cambridge, Massachusetts, United States of America

* atte.eskelinen@uef.fi

## Abstract

Degenerative musculoskeletal diseases like osteoarthritis can be initiated by joint injury. Injurious overloading-induced mechanical straining of articular cartilage and subsequent biological responses may trigger cartilage degradation. One early sign of degradation is loss of aggrecan content which is potentially accelerated near chondral lesions under physiological loading. Yet, the mechanoinflammatory mechanisms explaining time-dependent degradation in regions with disparate mechanical loading are unclear and challenging to assess with experiments alone. Here, we developed computational models unraveling potential mechanisms behind aggrecan content adaptation in fibril-reinforced porohyperelastic cartilage after single injurious overloading (50% compressive strain magnitude, 100%/s strain rate) followed by physiological cyclic loading (15% strain, 1 Hz, haversine waveform). The simulated adaptation of aggrecan content was compared spatially and at several time points to tissue composition found in Safranin-O-stained sections of young bovine knee cartilage subjected to the same loading protocols. Incorporating mechanical strain-driven cell damage and downstream proteolytic enzyme release, fluid flow-driven aggrecan depletion, and fluid pressure-stimulated regulation of aggrecan biosynthesis, the models agreed with experiments and exhibited 14%-points greater near-lesion aggrecan loss after 12 days of physiological loading compared to without loading. The near-lesion aggrecan loss was driven by fluid flow and proteolytic aggrecanase activity, while chondroprotective pro-anabolic responses (increased aggrecan biosynthesis) were prominent in the deeper tissue despite damaged superficial layer. This significant advancement in mechanistic understanding incorporated into cartilage adaptation model can help in development and guidance of personalized therapies, such as rehabilitation protocols and tissue-engineered constructs.

**Data availability statement:** The generated and analyzed data, metadata, codes, and models underlying this research are within the manuscript, Supporting Information files, and accessible via Finnish Fairdata research services: https://etsin. fairdata.fi/dataset/8613c535-a535-42cf-9b4d-c14b1a57723c, DOI: https://doi.org/10.23729/ fd-8c55f09b-d630-38db-ba0e-9bf8ac678018.

**Funding:** This work was supported by the Doctoral Programme in Science, Technology and Computing (LUMETO) of the University of Eastern Finland to JPK; the Research Council of Finland grant numbers 354916 to PT and 363459 to RKK; Strategic Funding of the University of Eastern Finland to PT; the Sigrid Jusélius Foundation to RKK; the Instrumentarium Science Foundation to ASAE; the Finnish Cultural Foundation, grant 00230091 to AE; the Maire Lisko Foundation to PT; the Päivikki and Sakari Sohlberg Foundation, grant 240074 to PT; the Novo Nordisk Foundation, grant NNF21OC0065373 to RKK. This project has received funding from the European Union's Horizon 2020 research and innovation programme under the Marie Skłodowska-Curie grant agreement Nos. 702586 to CF and 101108335 to AE. This work was supported by the Finnish Ministry of Education and Culture's Pilot for Doctoral Programmes (Pilot project Mathematics of Sensing, Imaging and Modelling; partial salary for ASAE). The funders had no role in study design, data collection and analysis, decision to publish, or preparation of the manuscript.

**Competing interests:** The authors have declared that no competing interests exist.

## Author summary

Post-traumatic osteoarthritis is an incurable musculoskeletal disease in which articular cartilage degenerates over time. One of the earliest signs of cartilage tissue degeneration is loss of its aggrecan content. Although both immobilization and excessive cartilage loading are known to cause aggrecan loss, the underlying cell-mediated inflammatory responses to loading — referred to as "mechanoinflammation" — remain unclear. Understanding mechanoinflammation necessitates methodological development which complements carefully designed experiments with state-of-the-art computational models incorporating cell-mediated mechanisms. Such biotechnological development presented here can be used to not only investigate cellular damage and tissue adaptation over time in differently loaded cartilage regions, but also to estimate the underlying, challenging-to-measure mechanical shear strains and fluid pressurization. Therefore, the presented computational framework of mechanoinflammation contributes to the development of models and computer-aided tools aimed at estimating and limiting osteoarthritis progression, while also being adoptable to research of other diseases.

## 1 Introduction

Post-traumatic osteoarthritis (PTOA) is a degenerative musculoskeletal disorder that develops over time following an injury to the knee joint. Injurious overloading and damage to articular cartilage are common in sports, rendering PTOA the predominant osteoarthritis phenotype in young active population [1,2]. In addition to visible tissue damage, mechanical insults and excessive mechanical loading may activate cell receptors and signaling pathways or even harm the cartilage cells which respond to this insult with catabolic enzyme production [3–5]. Defined as mechanoinflammation [6], this cascade leads to compositional changes such as loss of cartilage aggrecan content [7,8], an early sign of PTOA. However, it is yet not understood in which tissue regions and to what extent the underlying mechanoinflammatory mechanisms affect aggrecan content adaptation over time. Since evaluating the impact of different pathomechanisms experimentally is challenging, efforts could be reinforced by computational modeling to identify effective therapeutic solutions in halting PTOA progression.

The disruption of tissue homeostasis in early-stage PTOA has been studied experimentally using *ex vivo* cartilage explant cultures [5,7–14]. Following a controlled injurious overloading in (un)confined compression [15,16] or drop-tower setups [10,17], tissue-level catabolic responses such as substantial decreases in cell viability [5,7,18], extracellular matrix (ECM) proteoglycan content (PG, such as aggrecan) [8,13,14,19], and biosynthesis of new aggrecan [9,13,20] have been reported within the first 1–3 days. The injurious compression can excessively deform and shear the ECM [18] and increase the hydrostatic pressure rapidly to hyper-physiological levels [21–23]. Depending on the cellular microenvironment and the level of stress

or strain (rate) [24–28], cells may stay viable, undergo phenotypical changes (such as hypertrophy [29]), respond by exhibiting different types of cell damage (mitochondrial dysfunction and oxidative stress [10,12]), or die (apoptosis, necrosis) [18,28,30–32]. Cell damage alters cellular function that entails upregulation of proteolytic activity, *e.g.*, release of aggrecan-degrading aggrecanases. Aggrecanase activity has been detected *ex vivo* [4,13,33,34] and aggrecanase-related loss of aggrecan has been observed *in vivo* [35] within 1–2 days following injury.

Cyclic loading at normal physiological levels is beneficial for overall cartilage health. It promotes moderately varying fluid pressure over time, interstitial fluid flow, and solute transport [36,37], resulting in anabolic responses and accelerated aggrecan biosynthesis compared to static loading in explant cultures [38–42] and *in vivo* [43]. Yet, cyclic loading can also have locally degradative effects near lesions, where tissue (shear) stress/strain can rise to excessive levels, resulting in cell damage and subsequent aggrecan loss [7,14,44–46]. Physiological loading may also promote the transport of aggrecan fragments through damaged lesion surfaces via fluid flow [7,47].

Computational cartilage adaptation models have been developed to gain an understanding of the link between difficult-to-measure intra-tissue mechanical responses and experimental tissue alterations. Previous mechanistic modeling frameworks with either injurious [3,48] or cyclic loading [7,49] have connected loss of matrix proteins (PGs, collagen) to shear or compressive strain (rate) and tensile stress [7,26,27,50]. These models also implemented the decrease of cartilage load-bearing capability by iteratively updating permeability, matrix stiffness, or swelling/osmotic properties [27,50–52]. While the number of models incorporating cell-level responses [48,53,54], chemo–mechano–biological insights [3], and possible pro-anabolic effects [3,52] is currently limited, they provide important advancements in numerical estimation of mechanoinflammation. However, understanding how tissue-level mechanical loading alters cell viability/function and how that reflects on experimentally observed, spatio-temporal adaptation of aggrecan content is lacking.

Our aim was to develop a mechanistic modeling framework to explain localized and time-dependent (12-day) aggrecan loss in young bovine knee cartilage predisposed to injurious loading (INJ), physiological cyclic loading (CL), and their combination (INJ+CL). The major novel aspect the framework is equipped with is the estimates of cartilage cell responses to PTOA-triggering mechanical loading, and how the ensuing cell-driven mechanoinflammation regulates spatial aggrecan content in the early disease stages. Specifically, the framework utilizes time-dependent partial differential equations instead of an iterative approach with arbitrary time between model iterations. Intrigued by previous explant culture studies reporting depth-wise aggrecan loss and its worsening near chondral lesions with INJ+CL [7,14] (Fig 1A and 1B), we hypothesized that CL would induce heterogeneous strain and fluid flow fields with functional importance for cartilage cell and aggrecan distributions [22,32]. To replicate experimental aggrecan loss, we built control, INJ, CL, and INJ+CL models in a step-by-step fashion, adding mechanoinflammatory cartilage adaptation mechanisms as necessary (cell damage with aggrecanase release, fluid flow-driven depletion of aggrecan, fluid pressure-stimulated acceleration of aggrecan biosynthesis; Fig 1C), and performing sensitivity analyses.

## 2 Results

### 2.1 Cyclic loading accelerates experimental aggrecan loss near lesions

In the experiments, the aggrecan content within 50 μm from lesion edges was significantly lower than away from lesions on the day of injury ($p = 0.002$, linear mixed effects (LME) model with Bonferroni adjustment, Fig 1B). On day 12, the aggrecan contents near and away from lesions in the INJ group were similar ($p = 0.530$). In the INJ+CL group, the near-lesion aggrecan content was significantly lower than away from lesions on day 12 ($p < 0.001$, Fig 1B).

### 2.2 Computational models of mechanoinflammation reproduced the injury-related through-depth aggrecan loss as well as the locally detrimental and beneficial effects of physiological loading

After implementing the initial day-0 aggrecan content (Fig 2A), the computational model with intact geometry was first calibrated using day-12 depth-wise aggrecan content data from free-swelling control (CTRL) conditions. This model

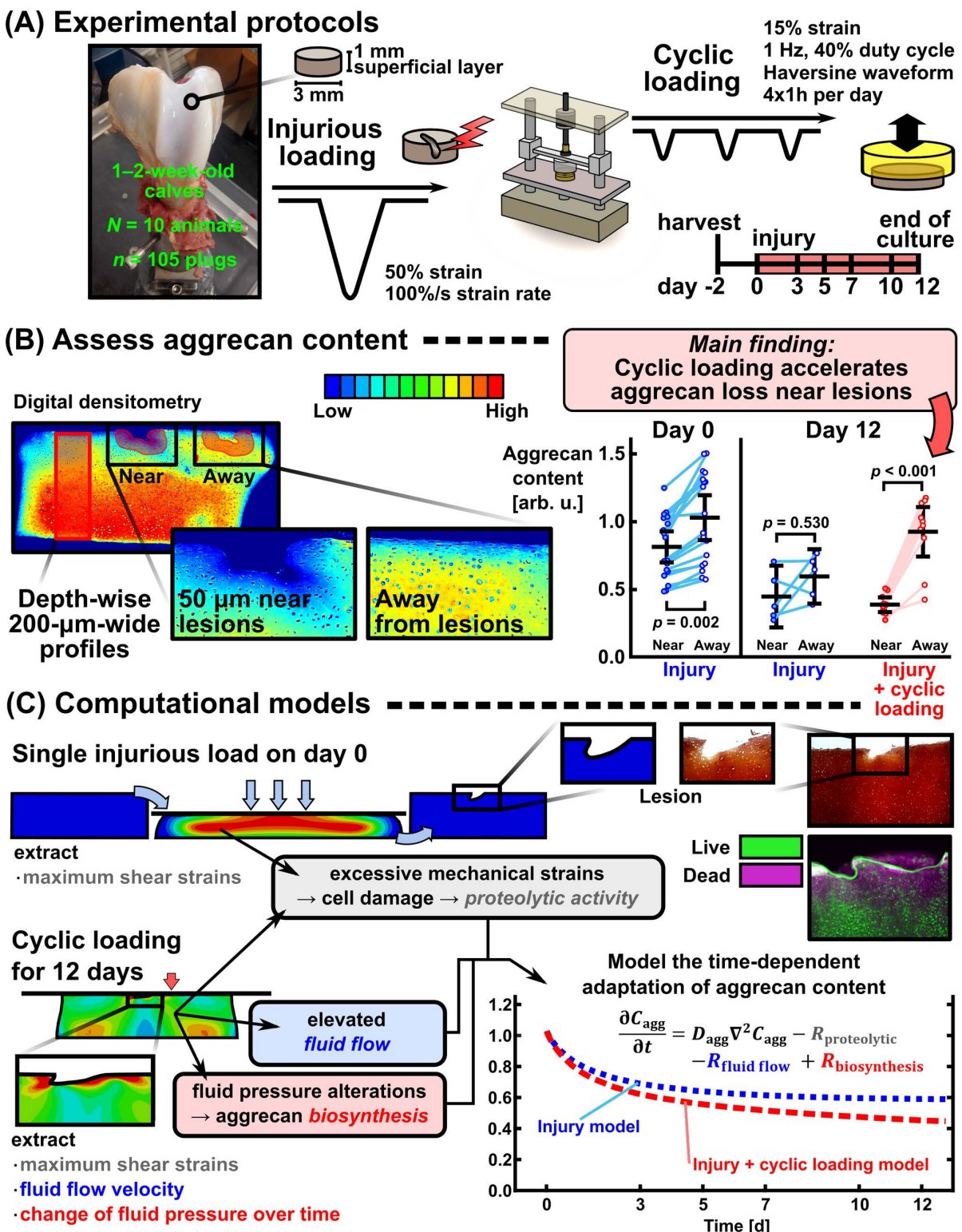

**Fig 1. Study workflow.** (A) In the experiments, cartilage explants were subjected to single injurious load–unload cycle followed by up to 12 days of physiological cyclic loading. (B) Cartilage aggrecan content was assessed from Safranin-O-stained sections with digital densitometry in depth-wise profiles and in regions near and away from chondral lesions. The plots with near and away from lesion data represent mean ± 95% confidence intervals, and

linear mixed effects model with Bonferroni adjustment. The day 0 values are pooled from two previous datasets [7,14]. (C) Computational finite element models included single injurious loading followed by physiological cyclic loading. After implementing three time-dependent, mechanobiological cartilage adaptation mechanisms (proteolytic activity upregulated by damaged cells in highly sheared regions, fluid flow-driven transport of aggrecan fragments out of tissue, and accelerated aggrecan biosynthesis due to fluid pressure changes over time), the injury and cyclic loading model replicated accelerated aggrecan loss near lesions compared to away from lesions, as observed in the experiments. Lesion formation was not modeled in this study.

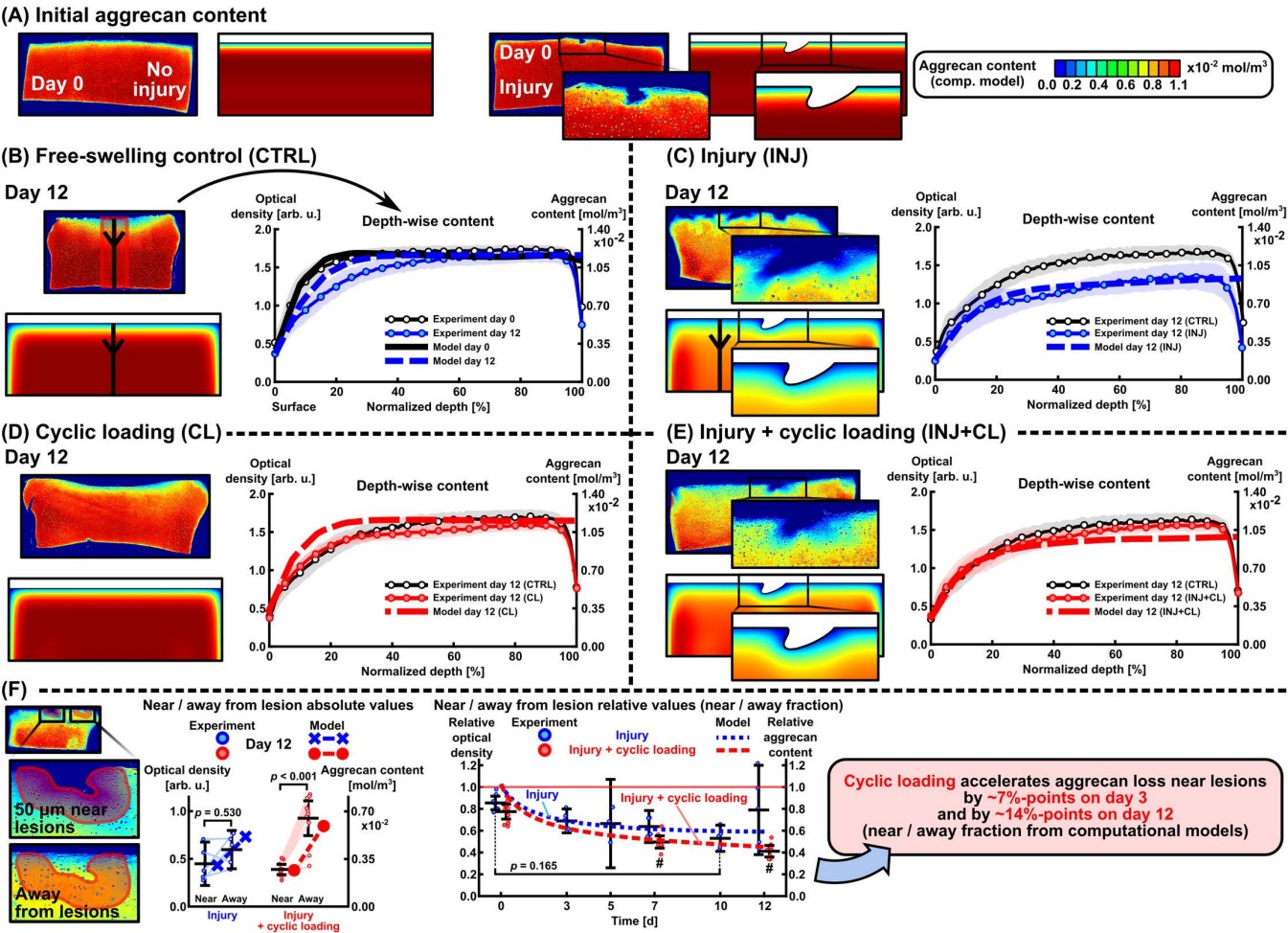

**Fig 2. The calibrated computational model reproduced the experimental findings of aggrecan content adaptation depth-wise, and near and away from chondral lesions over 0–12 days.** (A) Initial aggrecan distribution at the beginning of the culture and simulation. Depth-wise aggrecan content away from lesions in (B) free-swelling control, (C) injurious loading, (D) cyclic loading, and (E) injurious and cyclic loading experimental groups and computational models. (F) Numerical estimates of near and away from lesion aggrecan contents agreed with experiments in absolute (left panel) and relative terms (right panel; near/away fraction; value <1 means more aggrecan lost near lesions compared to away). The plots represent mean ± 95% confidence intervals, and linear mixed effects model with Bonferroni adjustment. # marks $p < 0.001$ compared to day 0 (right panel).

included free diffusion of aggrecan out of the tissue, resulting in an ~8–25% decrease in aggrecan content in contrast to ~10–30% mean decrease in the experiment at normalized depths of 0–40% (0% = surface, 100% = bottom of the cartilage explant) on day 12 compared to day 0 (Fig 2B). The INJ model included a single injurious load–unload cycle; in regions of excessive shear strain at the time of maximum compression, 'healthy' cells were assumed to switch their phenotype to

'damaged' cell (only these two cell populations were implemented; see Section 4.6, Tables 1 and B in S1 Text). Subsequently, the damaged cells released aggrecanases, promoting enzymatic activity for 12 days. This approach captured the substantial decrease in depth-wise aggrecan content in INJ compared to CTRL [7,14] (Fig 2B and 2C). The CL models with physiological loading for 12 days included the following mechanisms: shear strain-driven cell damage (and, thus, localized enzymatic activity), depletion of aggrecan fragments out of the tissue in regions of high fluid flow velocity, and accelerated biosynthesis of new aggrecan molecules in regions of moderate fluid pressure changes over time. The resulting depth-wise aggrecan contents in intact regions especially at normalized depths of 0–40% were higher in both the CL and INJ + CL models compared to the INJ, as observed in the experiments (Fig 2D and 2E).

The near-lesion aggrecan content was lower than the away-from-lesion content in both the INJ and INJ + CL models on day 12 (Fig 2F, left panel). The difference between near and away aggrecan contents was greater in the INJ + CL model $(0.21 \cdot 10^{-2}$ mol $\cdot$ m$^{-3})$ than in the INJ model $(0.32 \cdot 10^{-2}$ mol $\cdot$ m$^{-3})$ on day 12, consistent with the experiments (Fig 2F, left panel). Over time, the relative near/away from lesion aggrecan content decreased more rapidly in the INJ + CL compared to the INJ model. For the INJ model, the relative aggrecan content was 0.689 on day 3 (experiment: $0.690 \pm 0.110$ (mean±95% confidence interval)), 0.614 on day 7 $(0.638 \pm 0.147)$, 0.596 on day 10 $(0.530 \pm 0.120, p = 0.165$ compared to day 0 $(0.855 \pm 0.062))$, and 0.591 on day 12 $(0.790 \pm 0.410,$ Fig 2F, right panel). For the INJ + CL model, the relative aggrecan content was decreased from 0.622 on day 3 to 0.516 on day 7 (experiment: $0.496 \pm 0.056, p < 0.001$ compared to day 0 $(0.775 \pm 0.068))$ and to 0.452 on day 12 (experiment: $0.411 \pm 0.053, p < 0.001$ compared to day 0, Fig 2F, right panel). Cyclic loading accelerated aggrecan loss by 6.7%-points on day 3 (near vs. away fraction INJ: 0.689, INJ + CL: 0.622) and by 13.9%-points on day 12 (INJ: 0.591, INJ + CL: 0.452, Fig 2F, right panel).

## 2.3 Injurious loading may increase catabolic cell activity via excessive mechanical shear strains and rapid fluid pressure changes

The modeled injurious loading triggered whole-thickness loss of healthy cells [7,13–15,55,56] on day 0, observed as ~30–45% increase in non-healthy (damaged) cells [5,53,55] in the computational INJ model (Fig 3A, right panel). Various literature-supported biomechanical mechanisms for triggering the injury-associated cell damage and subsequent catabolic cell activity (aggrecanase release) were investigated [3,22,45,55–58]: maximum shear strain (damage initiation threshold: 40% [47,55,59], Fig 3A and 3B), rapid change of fluid pressure over the time of injurious compression (threshold: 80 MPa $\cdot$ s$^{-1}$ which over 0.5 s of ramping up the INJ loading corresponds to peak pressure in the range of ~40 MPa [22,58,60,61], Fig 3C), and compressive (logarithmic) axial strain (damage initiation threshold: 40% [47,55,59], Fig 3D). All these mechanisms resulted in a substantial decrease in whole-thickness aggrecan content via aggrecanases over 12 days. The simulated aggrecan content was similar between the maximum shear strain-driven (coordinate system-independent strain measure) and fluid pressure-driven models, both estimating lower aggrecan content compared to the axial strain-driven model (coordinate system-dependent strain measure, Fig 3E, left and middle panels). All three models were consistent with the experimental near/away from lesion aggrecan content fractions (day 7 experiment: $0.638 \pm 0.147$; shear strain model 0.614; fluid pressure model 0.627; axial strain model 0.606, Fig 3E, right panel).

Sensitivity analysis of the maximum shear strain-driven INJ model revealed that the threshold for shear strain-driven cellular damage $\varepsilon_{dmg,max}$ (in regions exceeding this threshold, fraction $k_{inj} = 0.45$ of healthy cells were considered damaged cells) had a minor effect on the absolute aggrecan content on day 12 (largest difference: model with $\varepsilon_{dmg,max} = 200\%$ resulted in 4.6% higher near-lesion aggrecan content than model with $\varepsilon_{dmg,max} = 100\%$, Fig 4A, middle panel). Increasing the values of parameters describing the amount of damaged cells $k_{inj}$ and aggrecanase release $k_{aga}$ decreased the aggrecan content considerably on day 12 ($k_{inj} = 60\%$ resulted in 33.5% lower near-lesion aggrecan content than $k_{inj} = 20\%$, Fig 4B; $k_{aga} = 0.375 \cdot 10^{-21}$ mol resulted in 26.0% lower near-lesion aggrecan content than $k_{aga} = 0.125 \cdot 10^{-21}$ mol, Fig 4C). These three parameters ($\varepsilon_{dmg,max}, k_{inj},$ and $k_{aga}$) had negligible effect on the relative near/away from lesion aggrecan content over time (maximum difference 3.2%-points in models with different $k_{aga}$ on day 12, Fig 4C, right panel). Selecting

**Table 1. Summary of the mechanoinflammatory equations and parameters in the four different models.** For the full model description of the cell damage and aggrecan loss, see Section 4.6 and Section C in <u>S1 Text</u>. CTRL = free-swelling control, INJ = injurious loading, CL = cyclic loading. Variable $z$ is normalized depth in cartilage ($z = 0$ surface, $z = 1$ bottom) and $H = 1$ mm is cartilage thickness.

| Variable/Parameter | Expression/Value | Description/Ref. |
|---|---|---|
| **Damaged cells** $C_{\text{cell,damaged}}$ **[cells·m$^{-3}$]** | | |
| In CTRL model | $C_{\text{cell,damaged}} = 0$ | No damaged cells [62] |
| In INJ model | $C_{\text{cell,damaged}} = C_{\text{cell,damaged,init}} = k_{\text{inj}}\, f_{\text{dmg}}(\varepsilon)\, C_{\text{cell,healthy,init}}$ | Eq. (2) [53,75]. On day 0, healthy cells turn damaged as per damage function $f_{\text{dmg}}(\varepsilon)$ in Eq. (1), with shear strain thresholds $\varepsilon_{\text{dmg,init}} = 40\%$ and $\varepsilon_{\text{dmg,max}} = 150\%$ [47,55,59]. Cell damage is linked to proteolytic activity via Eqs. (5) and (6) |
| In CL model | $\frac{\partial C_{\text{cell,damaged}}}{\partial t} = k_{\text{cl}}\, f_{\text{dmg}}(\varepsilon)\, C_{\text{cell,healthy}}$, with $C_{\text{cell,damaged,init}} = 0$ | Eqs. (4) and (S14). Rate of strain-driven cell damage during cyclic loading, without initial cell damage |
| In INJ+CL model | $\frac{\partial C_{\text{cell,damaged}}}{\partial t} = k_{\text{cl}}\, f_{\text{dmg}}(\varepsilon)\, C_{\text{cell,healthy}}$, with $C_{\text{cell,damaged,init}} = k_{\text{inj}}\, f_{\text{dmg}}(\varepsilon)\, C_{\text{cell,healthy,init}}$ | Eqs. (4) and (S14). Rate of strain-driven cell damage during cyclic loading, with initial cell damage from injurious loading of day 0 |
| **Aggrecan loss reaction terms** **in** Eq. (7) **[mol·s$^{-1}$·m$^{-3}$]** | | |
| $R_{\text{proteolytic}}$ in CTRL | 0 | No proteolytic activity |
| $R_{\text{proteolytic}}$ in CL, INJ, INJ+CL | $k_3 C_{\text{aga}} \frac{C_{\text{agg}}}{C_{\text{aga}} + K_{\text{m,aga}}}$ | Eq. (S17) [62]. Rate of change of aggrecan concentration $C_{\text{agg}}$ due to aggrecanases $C_{\text{aga}}$ |
| $R_{\text{fluid flow}}$ in CTRL, INJ | 0 | No fluid flow-driven aggrecan depletion in the absence cyclic loading over 12 days |
| $R_{\text{fluid flow}}$ in CL, INJ+CL | $k_{\text{cl,fl}}\, f_{\text{dmg}}(v)\, C_{\text{agg}}$ | Eq. (S18). Rate of change of aggrecan concentration $C_{\text{agg}}$ due to elevated fluid flow as per damage function $f_{\text{dmg}}(v)$, with similar form as Eq. (1), and fluid velocity thresholds $v_{\text{dmg,init}} = 0.08$ mm · s$^{-1}$ and $v_{\text{dmg,max}} = 0.15$ mm · s$^{-1}$ [7,47] |
| $R_{\text{biosynthesis}}$ in CTRL, INJ | $P_{\text{ag}} \left(1 + 0.9\frac{1-z}{H}\right) C_{\text{cell,healthy}} \left(1 - \frac{C_{\text{agg}}}{C_{\text{agg,tar}}}\right)$ | Eq. (S20). Rate of change of aggrecan concentration $C_{\text{agg}}$ due to aggrecan biosynthesis, with homeostatic target aggrecan concentration $C_{\text{agg,tar}}$ [7,62] |
| $R_{\text{biosynthesis}}$ in CL, INJ+CL | $(1 + f_{\text{synth}}(\hat{p}))\, P_{\text{ag}}$ $\cdot \left(1 + 0.9\frac{1-z}{H}\right) C_{\text{cell,healthy}} \left(1 - \frac{C_{\text{agg}}}{C_{\text{agg,tar}}}\right)$ | Eq. (S20). Rate of change of aggrecan concentration $C_{\text{agg}}$ due to aggrecan biosynthesis accelerated due to CL as per normalized biosynthesis function $f_{\text{synth}}(\hat{p})$, with similar form as Eq. (1), and fluid pressure change thresholds $\hat{p}_{\text{synth,init}} = 20$ MPa · s$^{-1}$ and $\hat{p}_{\text{synth,max}} = 60$ MPa · s$^{-1}$ [22,58,63–65] |
| **Parameters** | | |
| $k_{\text{inj}}$ [-] | 0.45 | Eq. (2) [16,53]. Fraction of damaged cells after injury |
| $k_{\text{cl}}$ [s$^{-1}$] | $1.5 \cdot 10^{-6}$ | Eqs. (4), (S13), and (S14), model fit. Rate of strain-driven cell damage during cyclic loading |
| $k_3$ [s$^{-1}$] | 0.9 | Eq. (S17) [62]. Catalytic rate constant for aggrecanase to degrade aggrecan |
| $k_{\text{cl,fl}}$ [s$^{-1}$] | $1.5 \cdot 10^{-6}$ | Eq. (S18), model fit. Rate of fluid flow-driven aggrecan depletion |
| $K_{\text{m,aga}}$ [mol · m$^{-3}$] | $5.5 \cdot 10^{-5}$ | Eq. (S17) [62,95,96]. Michaelis constant for aggrecanase |
| $P_{\text{ag}}$ [mol · cell$^{-1}$ · s$^{-1}$] | $2.4 \cdot 10^{-22}$ | Eq. (S20) [62,97,98]. Basal aggrecan biosynthesis rate (from healthy cells) |

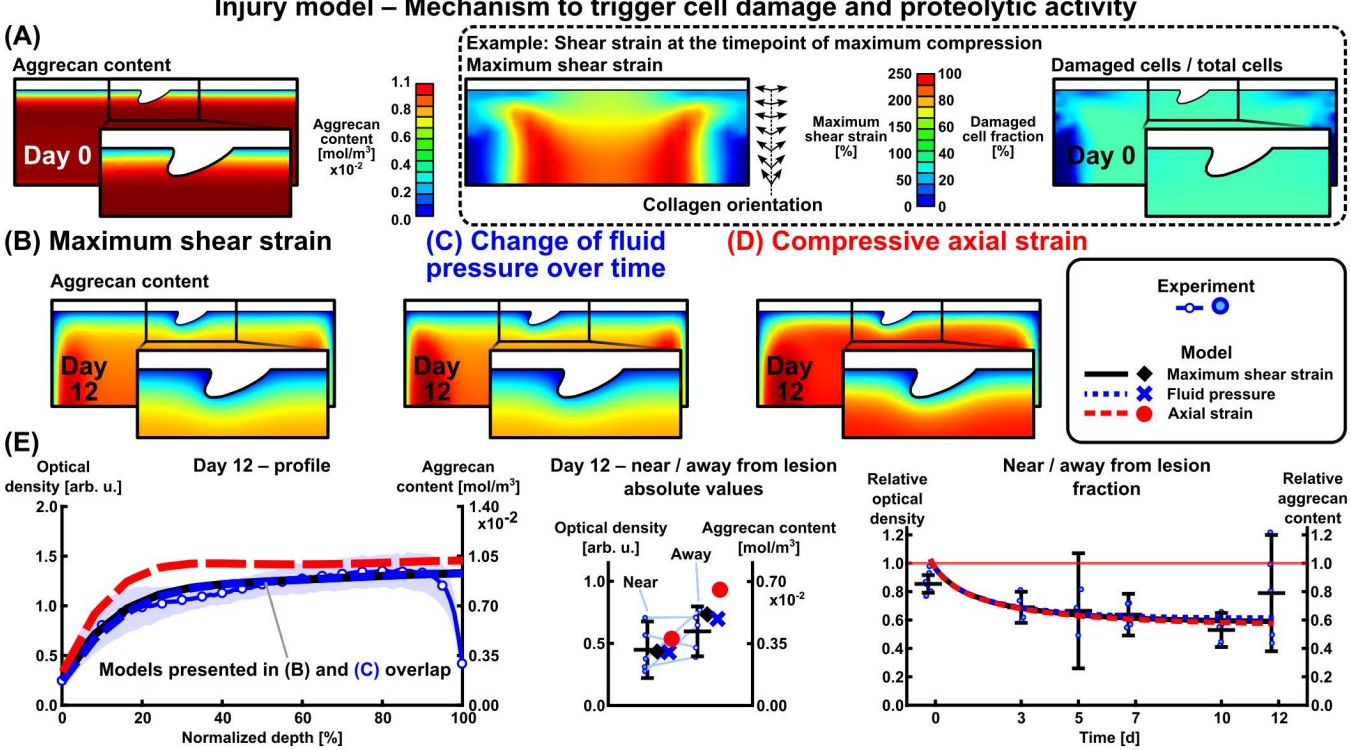

**Fig 3. Different mechanisms to trigger cell damage and subsequent proteolytic activity (aggrecanase release) in the injury-only model.** (A, B) Maximum shear strain (coordinate system-independent parameter) and (C) change of fluid pressure over time as mechanisms of cell damage resulted in similar estimates of aggrecan loss. (D) Compressive axial strain (coordinate system-dependent parameter) resulted in less aggrecan degradation than the maximum shear strain mechanism when all the model parameters were kept constant, as observed in (E) depth-wise, and near and away from lesion aggrecan contents. Maximum shear strain was selected for the rest of the study due to its coordinate system independence and more support from literature for the degradation thresholds as opposed to challenging-to-measure fluid pressures.

a higher initial depth-wise aggrecan content [62] compared to the current digital densitometry-based content [7] resulted in higher relative near/away aggrecan content over time (0.789 [62] *vs.* 0.591 [7] on day 12, Fig 4D, right panel).

## 2.4 Physiological loading promotes fluid flow-driven aggrecan loss near lesions and fluid pressure-stimulated acceleration of aggrecan biosynthesis in intact regions

The INJ+CL model was built step-by-step by first considering proteolytic activity only (localized cell damage from cyclic loading in addition to the initial injury-triggered catabolic cell activity, threshold of damage initiation 40% [47,55,59], Fig 5A). Due to mismatch with experimental near/away from lesion aggrecan content, additional cartilage adaptation mechanisms were investigated: elevated fluid flow velocity-related aggrecan depletion (threshold $0.08 \text{ mm} \cdot \text{s}^{-1}$ [7,47], Fig 5B) and acceleration of aggrecan biosynthesis by up to 100% from basal biosynthesis rate in regions where fluid pressure changes were within a healthy range ($20–60 \text{ MPa} \cdot \text{s}^{-1}$ which over 0.2s of ramping up the CL (1 Hz, 40% duty cycle) corresponds to peak pressures in the range of 4–12 MPa [22,58,63–65], Fig 5C). Implementing the fluid flow mechanism increased near-lesion aggrecan loss compared to the model with proteolytic activity only (Fig 5D, middle panel). Implementing the fluid pressure-stimulated acceleration of biosynthesis resulted in higher aggrecan content especially in away-from-lesion regions compared to models without accelerated biosynthesis (Fig 5D, left and middle panels). On day 12, the relative near/away from lesion aggrecan content was 0.566 with proteolytic activity only, decreasing to 0.500

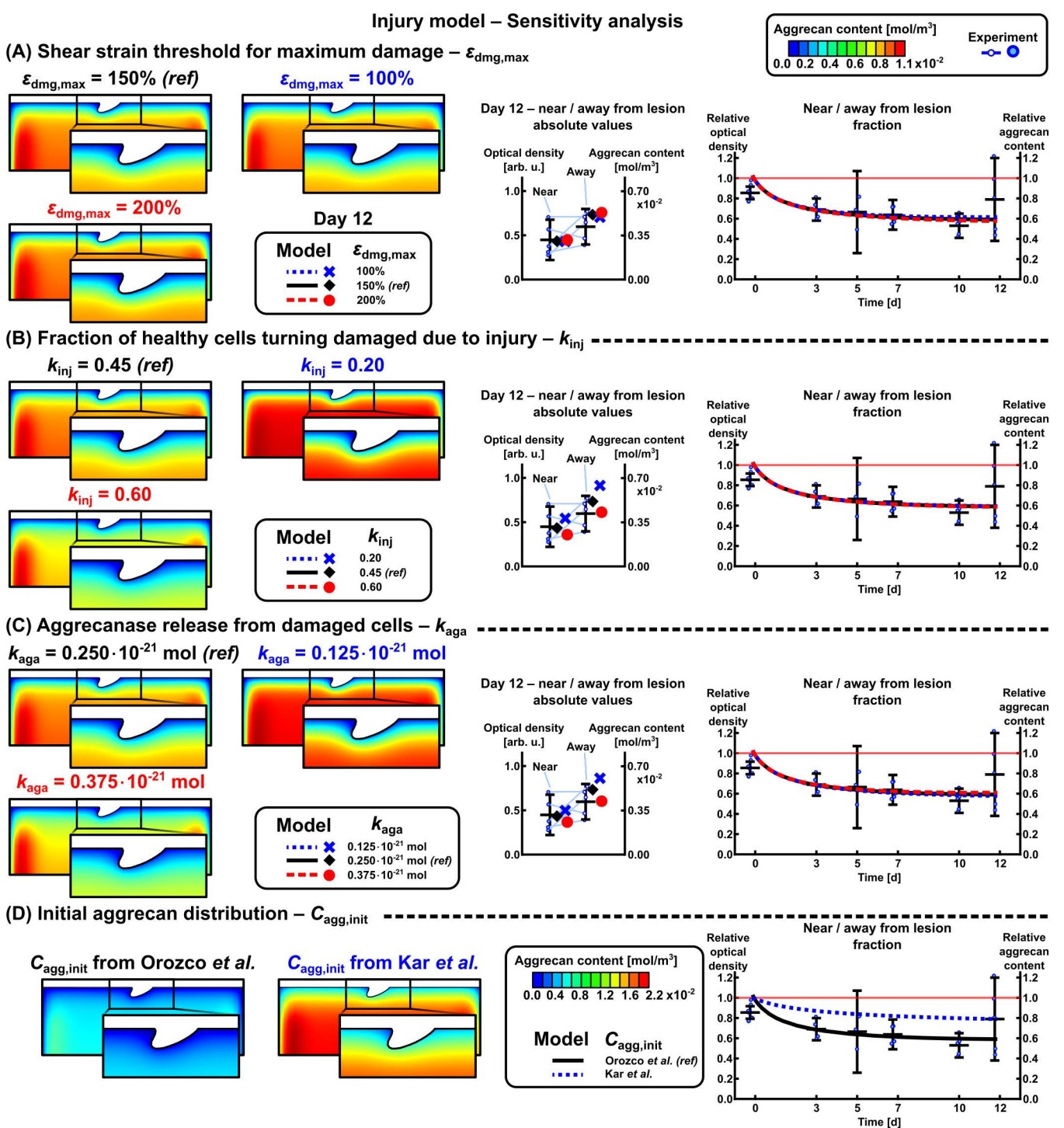

**Fig 4. Effect of selected parameters on the aggrecan loss in the injury model.** (A) Maximum shear strain threshold value for maximum damage, (B) fraction of healthy cells turning damaged due to injury, and (C) proteolytic activity (aggrecanase release) scaled the absolute amounts of localized aggrecan content, but their effect on relative near *vs.* away aggrecan contents was negligible. On the other hand, (D) initial aggrecan concentration affected the relative change in aggrecan content near *vs.* away from lesions. Note different colorbar scale in (D); the current study uses Orozco *et al.* [7] aggrecan distribution. *ref* = reference value selected to the models.

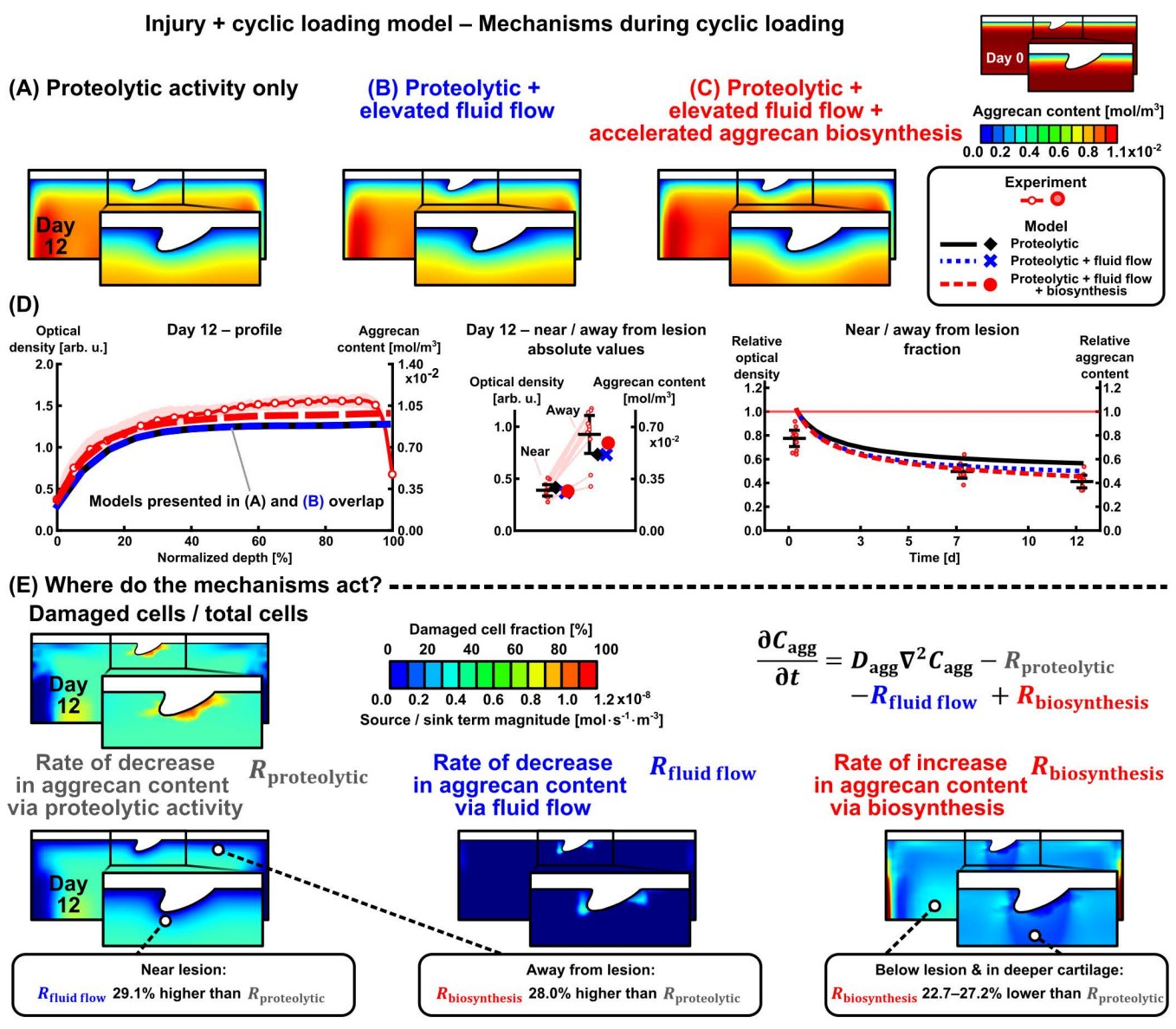

**Fig 5. Different mechanisms associated with cyclic loading in the injury and cyclic loading model.** (A) Only proteolytic activity due to damaged cells (triggered by maximum shear strain), (B) addition of the effect of elevated fluid flow flushing aggrecan fragments out of the tissue, and (C) addition of the effect of accelerated aggrecan biosynthesis when loading is within healthy limits (~change of fluid pressure over time). (D) Depth-wise, and near and away from lesion aggrecan contents. (E) Cell-driven proteolytic activity affects the whole geometry, elevated fluid flow acts near the lesion, and accelerated aggrecan biosynthesis plays a role in the bulk of the model excluding near- and below-lesion regions.

with proteolytic activity and fluid flow mechanisms combined, and to 0.452 with proteolytic activity, fluid flow, and fluid pressure-stimulated acceleration of aggrecan biosynthesis acting simultaneously (experiment: 0.411 ± 0.053, Fig 5D, right panel). Next, we used the model to numerically estimate the rate of aggrecan content increase/decrease associated to each of these mechanisms in localized areas on day 12 (magnitude of source/sink terms in Eq. (7)). Near the lesion, the effect of fluid flow was 29.1% higher than that of proteolytic activity (Fig 5E, left panel). Away from the lesion, the effect of fluid pressure-stimulated acceleration of aggrecan biosynthesis was 28.0% higher than the proteolytic effect (Fig 5E, middle panel). Below the lesion and in deeper cartilage away from the lesion, the effect of accelerated biosynthesis was

22.7–27.2% lower than the proteolytic effect ([Fig 5E](), right panel), with biosynthetic activity being lower below the lesion compared to away from the lesion.

From sensitivity analysis of the INJ + CL model, we highlight that increasing the rate of cell damage $k_{cl}$ and rate constant of aggrecan depletion due to fluid flow $k_{cl,fl}$ decreased aggrecan content near lesion on day 12 ($k_{cl} = 7.5 \cdot 10^{-6}$ s$^{-1}$ resulted in 5.4%-points lower near/away from lesion aggrecan content than $k_{cl} = 0.3 \cdot 10^{-6}$ s$^{-1}$, [Fig 6A](); $k_{cl,fl} = 7.5 \cdot 10^{-6}$ s$^{-1}$ resulted in 16.8%-points lower near/away from lesion aggrecan content than $k_{cl,fl} = 0.3 \cdot 10^{-6}$ s$^{-1}$, [Fig 6B]()). The threshold for accelerated aggrecan biosynthesis $\hat{p}_{synth,init}$ had negligible effect on near/away from lesion aggrecan content (maximum difference 3.2%-points, [Fig 6C]()), while lesion geometry affected by 10.9%-points on day 12 (confidence interval, [Fig 6D]()).

## 3 Discussion

### 3.1 Summary

We developed time-dependent computational models of cartilage mechanoinflammation to estimate changes in spatial aggrecan content under injurious and physiological cyclic loading (summarized in [Fig 7]()). The models replicated experimental findings from two prior explant culture studies of early-stage PTOA [7,14] in terms of depth-wise aggrecan loss (whole-thickness response), near and away from chondral lesion aggrecan loss (localized response), and near *vs.* away-from-lesion aggrecan content ratios (sample-specific difference in localized responses; [Fig 2]()). Implementing **(1)** proteolytic activity (aggrecanase release) from cells damaged due to excessive maximum shear strains (at day-0 injury and over the 12-day CL), **(2)** fluid flow flushing out aggrecan fragments, and **(3)** fluid pressure-stimulated acceleration of aggrecan biosynthesis in moderately loaded areas, resulted in ~7% and ~14% higher aggrecan loss near lesions with CL compared to without CL after 3 and 12 days from the injury, respectively ([Fig 2F]()). The non-localized bulk aggrecan losses observed in these experimental setups were consistent with prior findings from the literature [9,13], as discussed in detail elsewhere [7,14]. In line with our hypothesis, the mechanical response to CL in injured cartilage geometry was heterogeneous (depth-wise, near lesions). This resulted in the fluid flow effect dominating over proteolytic activity near the chondral lesions, accelerated aggrecan biosynthesis partially counteracting aggrecanase-driven aggrecan loss in the deep cartilage regions, and diminished biosynthetic activity below the lesions compared to regions away from them ([Fig 5E]()). Overall, this computational model offers a new basis for assessing biomechanics-driven spatio-temporal cartilage remodeling (both degradation and pro-anabolic responses).

### 3.2 Injury model

Simulated injurious loading resulted in cell damage throughout the depth of the explant ([Fig 3A](), right panel; [Fig 7A]()). This depth-wise cellular response aligns with reports of elevated oxidative stress and apoptosis/necrosis within the top 1 mm of injured cartilage [12,13,28,32]. We combined these cell fates into a "damaged cell phenotype" population; the fraction of damaged to total cells after injury (~30–45%, [Fig 3A]()) fell in the range of experimental studies with unconfined compression (20–50% [5], 40–55% [16]) and impact loading injury (40–65% [32]). The cell damage from injurious loading was assumed to occur immediately on day 0, although the timescales for different cell damage mechanisms vary from seconds to ~1 h to ~2 days [12,16,66]. The strain threshold $\varepsilon_{dmg,init} = 40\%$ for initiating damage was based on computational studies [47,59] but it is noteworthy that also lower shear strains of 7–12%, as observed under shear loading of immature cartilage, have been associated with cell damage/death [57,67]. This discrepancy originates from the modeling assumptions (*e.g.*, compressive loading conditions, material model including collagen orientation implemented in 2D) *vs.* the real tissue in the shear experiments [57,67]. To better accommodate the different shear strain–cell damage responses, a more probabilistic approach with probability density functions could be undertaken in the future. Here, using a lower strain threshold value would essentially result in a visually similar aggrecan distribution as when increasing aggrecanase production ([Fig 4C]()). Instead, the threshold $\varepsilon_{dmg,max}$ scaling maximum damage had negligible effect on aggrecan content ([Fig 4A]()).

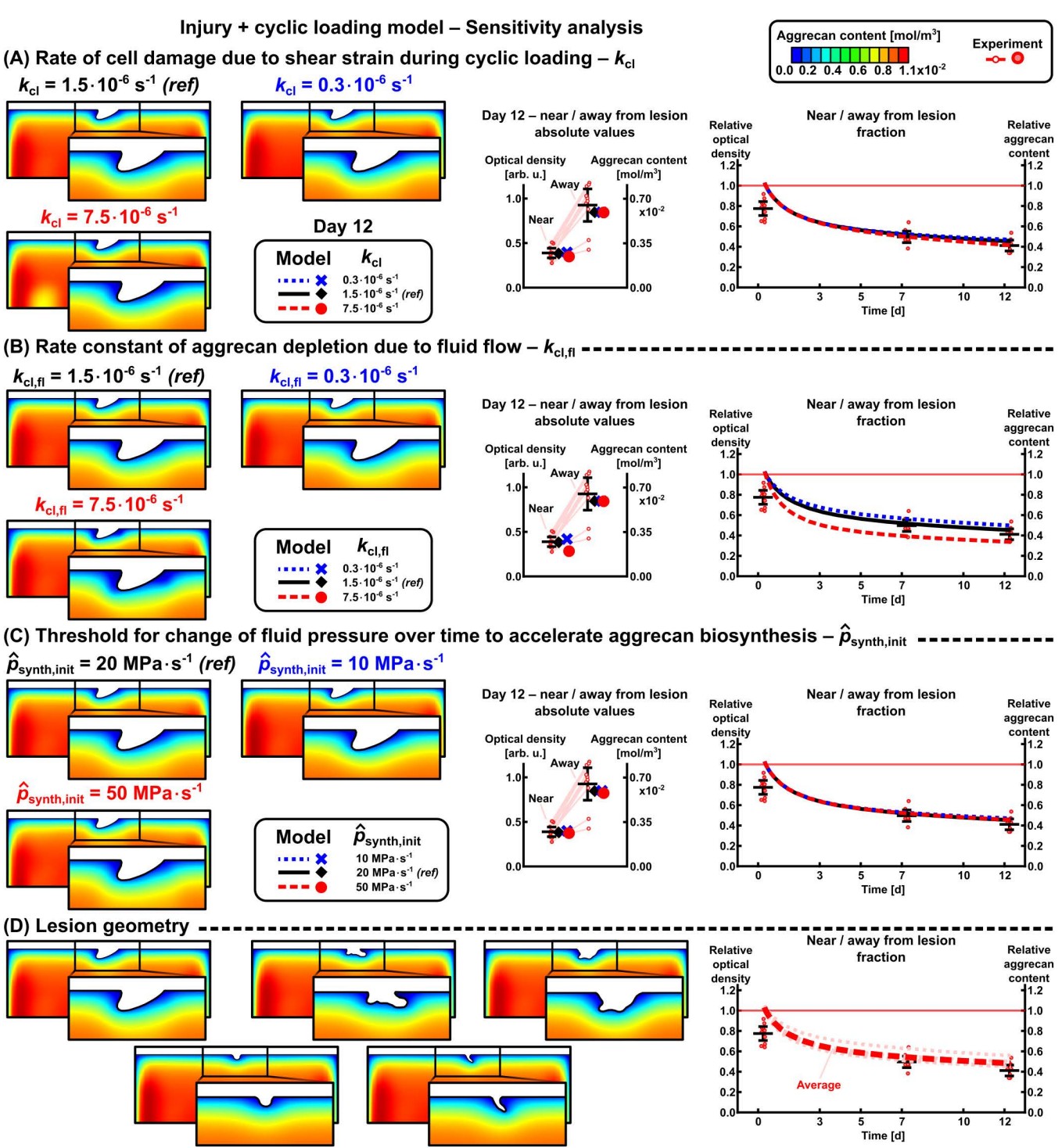

**Fig 6. Effect of selected parameters on the aggrecan content adaptation in the injury and cyclic loading model.** (A) Rate of cell damage (upregulating proteolytic activity) and (B) rate constant for aggrecan depletion due to fluid flow scaled the aggrecan loss especially near the lesion. (C) Fluid pressure-stimulated acceleration of aggrecan biosynthesis was present especially away from lesions, resulting in similar aggrecan content regardless of the chosen threshold. (D) Lesion geometry affected relative values of near vs. away from lesion aggrecan content for ~25%-points on day 12. *ref* = reference value selected to the models.

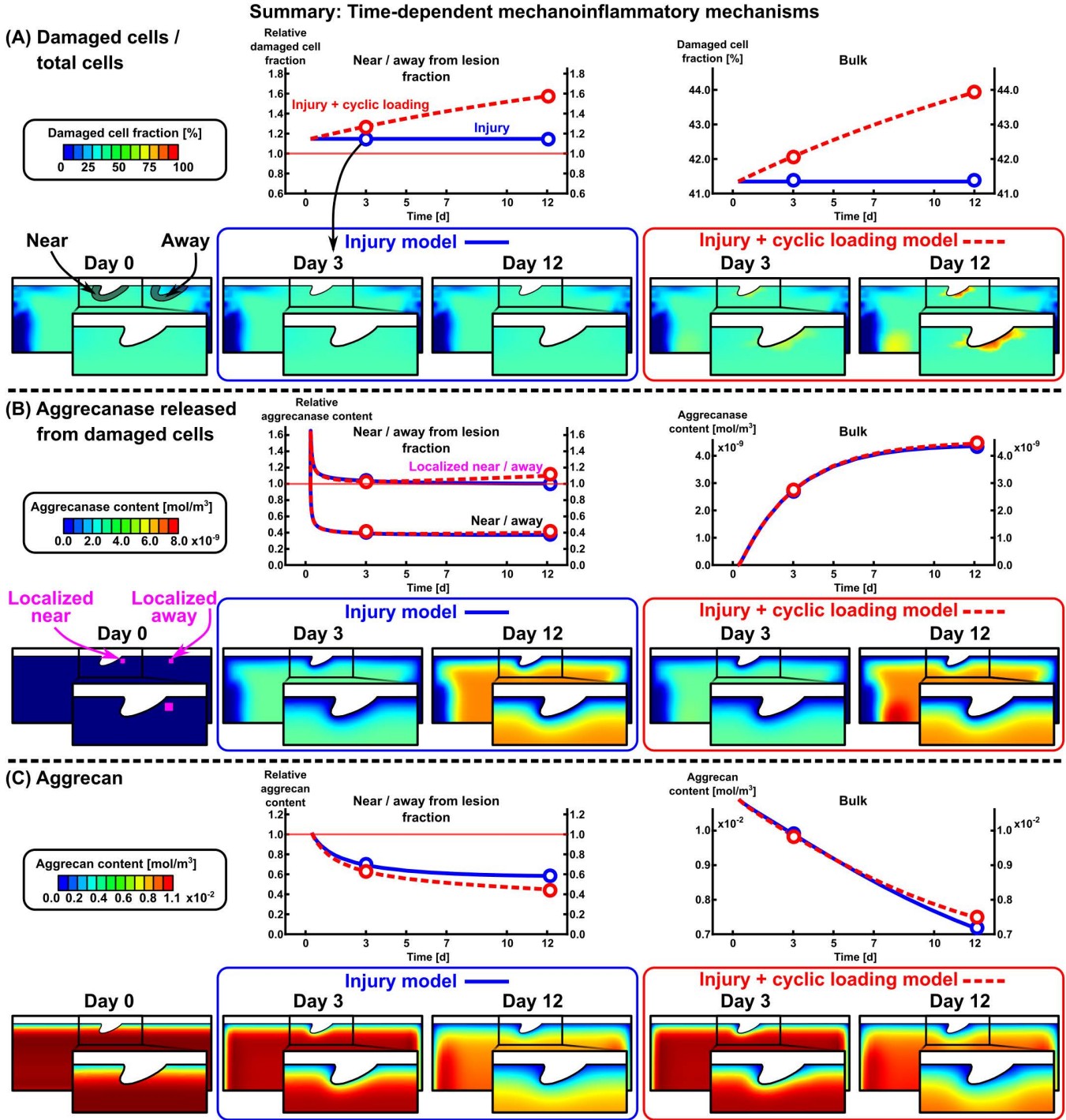

**Fig 7. Summary of the time-dependent mechanoinflammatory responses after injurious overloading.** (A) Cell damage was observed throughout the cartilage. Physiological cyclic loading (red dashed line) increased the fraction of damaged cells over time near the chondral lesion compared to injury-only (blue solid line). The cell damage maps as shown here were similar for the catabolic stimulus variable as per Eq. ([5]). (B) The aggrecanase content (released from damaged cells) increased rapidly in the first three days after injury. Cyclic loading further increased aggrecanase content but only in very localized areas near the lesion compared to away-from-lesion (magenta color; "localized near" region is from the superficial near-lesion region showing high cell damage, "localized away" is the same region translated to away from lesion). (C) The aggrecan content was lower near the lesion compared to away-from-lesion. The rate of aggrecan loss near the lesion increased when cyclic loading was involved, especially in the first three days.

By day 12, the average aggrecan content in the bulk tissue was higher in the injury and cyclic loading model compared to injury-only model, highlighting the overall beneficial effect of cyclic loading. Note that the aggrecan content was regulated not only by aggrecanases, but also by fluid velocity and aggrecan biosynthesis (see Fig 5E).

Localized release of aggrecanase from damaged cells was one of the key assumptions of the model. This cellular response can be detected locally with ADAMTS immunostaining [68], a potential method for finetuning the proteolytic parameter $k_{aga}$ that was found to strongly affect aggrecan content (Fig 4C). We based the value of $k_{aga}$ on the work of Kar *et al.* [62] — which focused on pro-inflammatory cytokine-driven aggrecan loss induced by interleukin (IL)-1 — with an assumption that stimulus for aggrecanase release would be similar for injurious loading (50% strain in 0.5s) and 1 ng/ml of IL-1 [13,14]. As a technical sidenote, $k_{aga}C_{cell,damaged}$ (Eqs. (5) and (S15)) was in a similar range as the equilibrium concentration of IL-1–IL-1-receptor complexes $C^*$ in [62]. A major part of immediate aggrecan loss is due to the microdamage from injury itself [4,69], which is not accounted for in the model, rendering the value of $k_{aga}$ an overestimate of the underlying damage mechanism.

Besides maximum shear strain (a coordinate system-independent strain measure), we also observed that coordinate system-dependent logarithmic axial strain (Fig 3D; comparable to minimum principal strain in simple geometries, Fig B in S1 Text) and fluid pressure change over time (Fig 3C) could be used to trigger cell damage and replicate the experimental findings. The axial strain is usable in simple geometries and loading conditions, while the shear strain, minimum principal strain, and fluid pressure could serve as viable computational biomarkers of cell damage in more complex joint-level models [70]. However, caution must be taken with the suggested and challenging-to-validate fluid pressure damage threshold (80 MPa $\cdot$ s$^{-1}$) serving as an upper limit where beneficial effects of loading turn detrimental [9,13,22].

### 3.3 Injury and cyclic loading model

Simulated 12-day CL on injured cartilage resulted in progressive and localized cell damage [66] but, as expected, without noticeable bulk damage in addition to the INJ only (Figs 3A, 5E, and 7A). The associated bulk proteolytic activity in the INJ only model increased rapidly over the first three days (Fig 7B), but with CL the released aggrecanase concentration increased only in very localized near-lesion areas compared to away-from-lesion (magenta highlight in Fig 7B). Thus, the proteolytic activity alone did not capture the observed aggrecan loss in the INJ + CL model (Fig 5D, right panel), necessitating the addition of the fluid flow mechanism to which aggrecan loss was sensitive near lesions (Figs 5B, 5E, and 6B). The selected fluid flow velocity threshold with reasonable model fit (0.08 mm $\cdot$ s$^{-1}$) was higher than a previous computationally estimated [7] value of 0.04 mm $\cdot$ s$^{-1}$. However, the results from this modeling framework suggest that fluid flow dominates over proteolytic activity in the depletion of aggrecan fragments near lesions during CL (Fig 5E), although this remains challenging to confirm experimentally. Modeling-wise this relatively low near-lesion proteolytic activity stems from using the same shear strain threshold in both INJ and CL (small near-lesion area with additional damaged cells) and setting aggrecanase concentration to zero on boundaries [62]. The latter results in high concentration gradient for aggrecanases (fast diffusion) and, thus, low aggrecanase content around the lesion (Fig 7B).

Pro-anabolic effects of CL were added as accelerated aggrecan biosynthesis to improve model fitting to experimental aggrecan loss in intact and deep regions (Fig 5D, left and middle panels). Increasing the basal aggrecan biosynthesis rate only in healthy cells by up to 100% (from experiments: 66–95% [71], 20–120% [40]; $^{35}$S-sulfate incorporation assays after similar CL protocols) partially counteracted the depth-wise proteolytic activity in deep and intact regions (Fig 5E, right panel) and resulted in higher bulk aggrecan content in the INJ + CL model compared to the INJ model on day 12 (Fig 7C). With this modeling framework, the aggrecan content showed low sensitivity to the suggested biosynthesis thresholds (10–50 MPa $\cdot$ s$^{-1}$, Fig 6C). Aggrecan biosynthesis was relatively low below the central lesion (Fig 5E), consistent with reports of decreasing hydrostatic pressure gradient, biosynthesis rate, and availability of solutes/nutrients from central to lateral regions during unconfined compression [3,72]. Alterations in aggrecan biosynthesis could also be

attributed to the activation of latent growth factors [3] and localized energy dissipation to the tissue [73], although these were not modeled here.

### 3.4 Limitations and future directions

A limitation of this computational model is the inclusion of only two cell populations, "healthy" and "damaged". Cartilage mechanoinflammation is far more complex, involving, *e.g.*, mechanical activation of ion channels (Piezo1/2, transient receptor potential vanilloid TRPV4) and downstream phenomena such as oxidative stress, mitochondrial dysfunction, apoptosis/necrosis, and cell hypertrophy [12,18,30,32,74]. The affected cell populations may orchestrate time-dependent reparative/remodeling responses to injury such as upregulating production of ADAMTS and tissue inhibitors of matrix metalloproteinases in surrounding viable cells (healthy and/or damaged) and, depending on the microenvironment, switching damaged cells to dead or back to healthy phenotype [4]. Modeling-wise the cellular variables and associated production/loss rates for different cell populations (as in Eq. (4)) may be implemented as linked to a certain, hypothesized mechanism or as basal sink/source terms when the real mechanism is not fully known. This approach has been adopted in previous cell–tissue-level models investigating cell death, hypertrophy, and oxidative stress, with comparison to experiments as far as the available data allow [3,53,54,75]. The challenge of estimating how the cell population concentrations evolve in different mechanoinflammatory environments may also be explored with intra-cellular signaling models [76], the linking of which to reaction–diffusion models remains to be established. While the basal cell production/loss terms might become more relevant in longer time scale PTOA models, their relevance in injury models of short time scale is uncertain. For these reasons, we chose to focus on replicating experimental findings with a minimal number of cell crosstalk-related parameters, which are explored elsewhere [3,48,52–54].

The model also omits the tissue responses to injury-related release and diffusion/advection of pro-inflammatory cytokines [13,49]. While exogenous cytokines have been included in previous models [3,53], their involvement introduces new difficult-to-validate parameters regarding healthy *vs.* damaged cell responses to cytokines (ADAMTS release, aggrecan biosynthesis) and relative magnitude of mechanics- *vs.* cytokine-driven catabolism. Justification and experimental comparison of these cell response parameters were out of scope of the current study.

Another limitation is the use of an idealized model geometry with perfectly flat and horizontal surface, and constant thickness, as opposed to experimental samples that may have had uneven surfaces and non-constant thicknesses (Fig 1B). This idealized model geometry has been suggested to reduce heterogeneity in strain fields during loading (and, thus, in simulated adaptation responses, Fig 7) when compared to models with sample-specific geometries [77]. Information on sample-specific geometry and material properties under high-rate loading, as well as corresponding numerical formulations for material behavior and damage initiation (*e.g.*, collagen failure), are all necessary for properly modeling lesion formation/propagation for instance with extended or cohesive finite element modeling (omitted in this study) [78,79]. Considering these simplifications, this study provides an estimate of the upper limit for the parameters of mechanical loading-related cell responses (peak biomechanical response used as input here instead of time-averaged estimates of the whole 12 days) and ECM remodeling in a single experimental setup.

While utilizing young bovine cartilage provides a repeatable setup in terms of mechanical and biological tissue properties, it must be acknowledged that the present numerical estimates of mechanoinflammatory processes — with some model parameters based on prior computational models when experimental evidence was unavailable — may not be fully translatable to more mature large animal or human cartilage. Moreover, the results are dependent on the experimental protocol; use of different loading amplitudes and frequencies, sample-specific geometries, and larger sample sizes could improve the robustness of the model parameters across different loading scenarios. Therefore, new explant culture/bioreactor studies should be conducted, not only with immature cartilage but also with mature (human cadaver) cartilage, in order to fully calibrate the numerical representation of the underlying mechanoinflammation. Another approach is organ-on-a-chip models with human cells, providing added control over desired mechanobiological processes compared to *ex vivo* models

[80,81]. To further promote the translational ability of the model to the knee joint-level, the simulated aggrecan depletion should be confirmed in joint-level mechanoinflammatory models (*e.g.*, comparison to quantitative MRI maps at baseline and several follow-up time points [59]) and compared against matrix biomarkers released to synovial fluid or systemic circulation [35,82]. Ultimately, such models could be utilized to design various rehabilitation exercises or tissue-engineered constructs to tackle disease progression.

Degradation softens cartilage, potentially leading to locally increasing strains over time. This has been captured in previous models focusing on collagen failure, fatigue, aggrecan depletion, and permeability increase [7,50] by using an iterative approach, where ABAQUS material model was updated in each iteration (arbitrary unit of time, necessitating the introduction of new parameters). Rigorous implementation of collagen failure to the current time-dependent mechanoin- flammatory model would necessitate use of biomechanical tests, collagen content maps, and estimates of matrix metal- loproteinase activity at several time points which is a topic for future work. During model development, we noticed a ~ 5% increase in maximum shear strains near the lesion in CL model with decreased aggrecan content (day 12 content of INJ + CL model), which would reduce aggrecan content approximately by <5% (Eq. (1), $k_{cl}$ in Fig 6A). While tissue soften- ing may not be a significant factor over 12 days, it becomes important over longer times in joint-level models.

### 3.5 Conclusions

By implementing several mechanoinflammatory cartilage adaptation mechanisms, the current modeling framework offers an avenue for studying the interplay between different loading scenarios, cartilage degradation and, as a relatively uncom- mon feature in the contemporary finite element modeling literature of osteoarthritis, the pro-anabolic responses in geome- tries with experimentally observed chondral lesions. The degradative and beneficial aspects of loading as presented here can be extended to knee joint level models to develop subject-specific osteoarthritis progression models [50] and evaluate the effects of physical rehabilitation schemes [70,83] to study cartilage damage for example after sports injuries. More generally, the modeling framework could also be developed further to simulate anti-catabolic and pro-anabolic therapeutic strategies [84]. Thus, the present study provides one piece to the multi-scale computational approaches aimed at helping limit PTOA progression.

## 4 Methods

### 4.1 Experimental groups and timeline

The experiments have been described in detail in two previous studies investigating localized aggrecan loss, which is observable already after 12 days in this experimental model of early-stage PTOA [7,14]. This study utilizes a subset of those data. Briefly, cartilage explants (thickness 1 mm, diameter 3 mm, $n = 105$) were harvested from patellofemoral grooves of 1–2-week-old calves on the day of slaughter ($N = 10$ animals, one knee per animal, Research 87 Inc., Boyl- ston, MA). After equilibration for two days in serum-free culture medium (marking day 0), the explants were divided into the following groups: free-swelling control (CTRL, $n = 19$, cultures terminated on days 0 and 12) [7], injurious loading only (INJ, $n = 54$, days 0 [7,14], 3, 5, 7, 10, and 12 [14]), cyclic loading only (CL, $n = 11$, day 12) [7], and injury and cyclic loading groups (INJ + CL, $n = 21$, days 7 and 12) [7].

### 4.2 Mechanical loading protocols

On day 0, explants in the INJ and INJ + CL groups were subjected to a single load–unload cycle of unconfined com- pression with 50% axial strain relative to the sample-specific thickness (strain rate 100%/s) within a custom-designed incubator-housed loading apparatus [9,13,14,85,86]. The resulting peak stresses ranged from 20–30 MPa (peak force per top surface area of the undeformed explant). This protocol has been demonstrated to decrease cell viability, especially near lesions [5,7,13,14], and to cause both bulk and localized loss of aggrecan content [8,13,14].

The CL and INJ + CL explants were subjected to cyclic loading protocol designed to mimic physiological loading and daily activities. After establishing repeatable contact with the explants with 10% compressive offset strain (five 2% axial strain ramp-and-hold increments, each increment followed by a six-minute stress-relaxation), the explants were subjected to cyclic unconfined compression with 15% axial strain (1 Hz, haversine waveform, 40% duty cycle, 1-h loading sessions followed by 5-h rest with the applied load removed; this protocol was repeated four times a day) in an incubator-housed cyclic loading apparatus [7,13,14]. Our choice to use haversine instead of sine waveform was motivated by physiological relevance, and the use of haversine waveform resulted in minimization of lift-off of the contact surface from cartilage surface during the active part of cyclic loading [39,40,87]. However, we cannot rule out the possibility of near-lift-off situations for some of the samples. This CL protocol has previously been shown to be non-detrimental for the bulk cartilage (negligible aggrecan loss [7], increased aggrecan biosynthesis [13,14]) with the potential to inhibit pro-catabolic responses in cartilage following mechanical injury [13].

### 4.3 Analysis of aggrecan content

Aggrecan content was estimated by measuring optical density (arbitrary units) in three 3-µm-thick, Safranin-O-stained cartilage sections per explant using digital densitometry (conventional light microscope, calibration with series of filters of known optical density, Nikon Microphot FXA, Nikon Inc., Tokyo, Japan, 4 × magnification, pixel size 1.23 µm) [14,88]. Injured explants with sections showing excessive tissue rupture or no lesions were excluded from the analysis (14/75 explants, 18.7%). For each section, optical density was measured in the following regions: **(1)** full-thickness depth-wise profiles (two 200-µm-wide regions, one from either side of lesion, averaged along the width and then depth-wise, Fig 1B), **(2)** near lesions (average value within 50 µm from lesion edges; mean lesion depth 135 ± 18 µm (95%CI), mean lesion width 204 ± 45 µm), **(3)** away from lesions (same analysis region shape as near lesion, translated to both sides of lesion and averaged; mean distances from lesion edges to away-from-lesion region edges 653 ± 97 µm, and from lesion center-points to away-from-lesion region edges 743 ± 101 µm). For **(4)** relative near *vs.* away from lesion fractions, the average optical density value near lesion was divided by the average away-from-lesion value. Lastly, each measure was averaged over the three sections per explant.

### 4.4 Statistical analyses

All statistical analyses were conducted using linear mixed effects (LME) models in IBM SPSS Statistics 29.0 (SPSS Inc., IBM Company, Armonk, NY, USA). Data points more than 1.5 times the interquartile range below the 25% quartile or above the 75% quartile were discarded as outliers. Different animals were considered as subjects with random intercepts, and treatment, time, and patellofemoral surface region where each explant was harvested from, as fixed factors (in the statistical analysis, the combined dataset had four surface regions from one study [14] and one region from the other [7]). Location-matching was done by harvesting explants for all treatment groups from each patellofemoral region. The covariance structure was set as the variance components structure (default). The significance level was set to α = 0.05, and results are reported with Bonferroni correction.

### 4.5 Computational models of mechanical loading

Two unconfined compression loading models were developed in ABAQUS (v. 2023, Dassault Systèmes Simulia Corp.) according to the experimental protocols: one for the initial single injurious compression on day 0 (50% strain in 0.5s; 100%/s) and another for the physiological cyclic loading (15% strain, with 40% duty cycle the maximum strain was obtained in 0.2s, two cycles modeled [7]). With cartilage modeled as a fibril-reinforced porohyperelastic swelling material incorporating depth-dependent compositional properties [7,89] (Section A and Table A in S1 Text), the injurious loading was subjected to intact geometry (peak stresses ~50 MPa when using material parameters found from literature with

which model was still converging, higher than ~30 MPa in experiments). Lesion formation or crack propagation [78] were not modeled. Instead, the conventional light microscopy images of the cartilage sections were imported to 3D Slicer (v. 5.4.0) [90] for manual segmentation of five representative lesion geometries of varying size and shape (five different plugs). The geometries were then imported to ABAQUS, where the sizes and shapes of the lesions were sketched with the spline tool on intact geometry (3-mm-wide by 1-mm-thick rectangle). These idealized intact and injured geometries were meshed in ABAQUS. Cyclic loading was subjected to intact (CL group) or injured geometry (INJ + CL group, five lesion geometries based on the cartilage sections, Fig 6D). All bottom surface nodes were fixed in vertical direction and in addition the middle bottom node was fixed in the horizontal direction, allowing for lateral expansion of the tissue. Fluid flow was allowed through the lateral and lesion surfaces (pore pressure = 0). Mesh convergence for both the injurious loading (Section B and Fig A in S1 Text) and cyclic loading models [7] was assured. The models were developed in 2D to facilitate the comparison of simulated and experimentally observed aggrecan loss in cartilage sections with lesions.

The mechanical loading models provided input parameters to the time-dependent mechanoinflammatory cartilage adaptation models described in Section 4.6 (see Section A in S1 Text for details on the parameters calculated from mechanical loading models). These included maximum shear strain as a trigger for day-0 cellular initial conditions in INJ (during model development, also fluid pressure change over time and logarithmic axial strain, Fig 3), and maximum shear strain, fluid flow, and fluid pressure change over time (promoting accelerated aggrecan biosynthesis) in the 12-day simulations of CL and INJ + CL (Fig 5 and Section C in S1 Text). The parameters were obtained at the timepoint of maximum compression; the parameter 'fluid pressure change over time' (rate of pressure buildup has been suggested to affect cellular responses [4]) was calculated between the beginning of the compression and the maximum compression (in ABAQUS, pore pressure change between two time points divided by the time period; over 0.5s in INJ, over 0.2s in CL and INJ + CL).

## 4.6 Computational models of mechanoinflammatory cartilage adaptation

The biomechanical model outputs from ABAQUS (such as, maximum shear strain $\varepsilon$) served as inputs for the mechanoinflammatory cartilage adaptation models with spatio-temporally evolving aggrecan concentration (COMSOL Multiphysics, v. 5.6., COMSOL AB, Stockholm, Sweden). The injurious loading was assumed to trigger initial cell damage on day 0, implemented with a normalized shear strain-driven cellular damage function $f_{dmg}(\varepsilon)$ [27,91] as

$$f_{dmg}(\varepsilon) = \begin{cases} 0, & \text{if } \varepsilon < \varepsilon_{dmg,init}, \\ \dfrac{\varepsilon_{dmg,max}}{\varepsilon} \dfrac{\varepsilon - \varepsilon_{dmg,init}}{\varepsilon_{dmg,max} - \varepsilon_{dmg,init}}, & \text{if } \varepsilon_{dmg,init} \leq \varepsilon \leq \varepsilon_{dmg,max}, \\ 1, & \text{if } \varepsilon > \varepsilon_{dmg,max}, \end{cases} \tag{1}$$

where thresholds for cell damage initiation $\varepsilon_{dmg,init} = 40\%$[47,55,59] and for maximum damage $\varepsilon_{dmg,max} = 150\%$ [55] (obtained from computational studies). Equations of this form were also used in the INJ models to investigate other potential cell damage triggers (excessive fluid pressure change over time and logarithmic compressive axial strain, Fig 3C and 3D and Table B in S1 Text). The damage function $f_{dmg}(\varepsilon)$ was used to convert part of the initial concentration of healthy cells $C_{cell,healthy,init}$ into damaged cells $C_{cell,damaged,init}$ on the day of injury [15,55] with the following initial condition [54]:

$$C_{cell,damaged,init} = k_{inj}\, f_{dmg}(\varepsilon)\, C_{cell,healthy,init}, \tag{2}$$

where $k_{inj} = 0.45$ is the maximum fraction of healthy cells turning to damaged phenotype in young bovine cartilage after injurious compression/impact [5,16,32,53] (Table 1). Due to the lack of suitable quantitative data, this modeling approach which involved only two cell populations aimed to represent the overall phenotypic switching from "healthy" to "damaged" cells without explicitly considering the underlying cell damage mechanisms separately (apoptosis [5,15,53], necrosis

[15,53,66], oxidative stress [12,53,71], mitochondrial dysfunction [92], hypertrophic state [3,93]). Depending on cellular microenvironment, the damage may be considered minor (altered cellular function) or major (cell death), with both scenarios associated with aggrecanase release either from the damaged cells themselves or from viable cells adjacent to damaged regions [4]. Therefore, as a generalization, we assumed that the modeled entity "damaged cells" would release aggrecanases (ADAMTS-4,5, a disintegrin and metalloproteinase with thrombospondin motifs) that diffuse within the cartilage ECM and eventually decrease the aggrecan content [53,68]. The cartilage adaptation was modeled with time-dependent diffusion–reaction equations over the 12 days following injury [49,62]:

$$\frac{\partial C_i}{\partial t} = D_i \nabla^2 C_i + \sum_j R_{i,j},$$

(3)

where $C_i$ is the time-dependent concentration of the species $i$ (healthy cells, damaged cells, aggrecan, aggrecanase), $D_i$ is effective diffusivity (0 for both cell populations), $t$ is time, and $R_{i,j}$ are the source (+)/sink (−) terms for species $i$ and mechanism $j$ (e.g., aggrecanase activity, aggrecan biosynthesis). The damaged cell population $C_{cell,damaged}$ evolved over time as

$$\frac{\partial C_{cell,damaged}}{\partial t} = k_{cl}\, f_{dmg}(\varepsilon)\, C_{cell,healthy},$$

(4)

where $k_{cl}$ is the rate of cells turning from healthy to damaged due to cyclic loading and $C_{cell,healthy}$ is the concentration of healthy cells. The presence of damaged cells gave rise to localized, time-dependent catabolic stimulus for aggrecanase release $S_{aga}$. The introduction of a stimulus variable — that has time-dependent rate coefficients to capture the time delay in mechanoinflammatory responses but no diffusion per se — was motivated by Kar et al. and experimental findings [4,69]. The latter includes observations that cell-driven enzymatic activity is delayed (in contrast to very early aggrecan loss due to structural microdamage from the mechanical insult) and increases after hours–days from injury [69] (delay due to trauma-related release of alarmins and damage-associated molecular patterns peaking in ~24h [4]).

$$\frac{\partial S_{aga}}{\partial t} = \alpha_{aga}\, (k_{aga} C_{cell,damaged} -\, S_{aga}),$$

(5)

where $\alpha_{aga}$ is rate constant for aggrecanase stimulus, and $k_{aga}$ is constant for aggrecanase release from damaged cells. The catabolic stimulus $S_{aga}$ was linked to the production of aggrecanases (that diffuse through the extracellular matrix) as

$$\frac{\partial C_{aga}}{\partial t} = D^*_{aga} e^{-d_1 C_{agg}} \nabla^2 C_{aga} + k_1 S_{aga} - k_2 C_{aga},$$

(6)

where $D^*_{aga}$ is diffusivity of aggrecanase, $d_1$ is a constant defining the dependency of aggrecanases' effective diffusivity on local aggrecan concentration ($D_{aga} = D^*_{aga} e^{-d_1 C_{agg}}$), $k_1$ is rate constant for generating aggrecanases, and $k_2$ is enzymatic loss of aggrecanases. Finally, the aggrecan concentration $C_{agg}$ evolved as:

$$\frac{\partial C_{agg}}{\partial t} = D_{agg} \nabla^2 C_{agg} - R_{proteolytic} - R_{fluid\ flow} + R_{biosynthesis},$$

(7)

where the proteolytic activity decreasing aggrecan content $R_{proteolytic}$ depended on the release of aggrecanases $C_{aga}$ from damaged cells (catabolic cell activity driven by both injury on day 0 and cyclic loading over 12 days; for the full equations see Table 1, and Section C and Table B in S1 Text). Elevated fluid flow-driven aggrecan depletion $R_{fluid\ flow}$

**Table 2. Parameters selected to the sensitivity analysis. Bolded values represent reference values.**

| Parameter | Values | Description | Ref. |
|---|---|---|---|
| $\varepsilon_{\text{dmg,max}}$[%] | 100, **150**, 200 | Shear strain threshold for maximum cell damage, Eq. (1) | [55] |
| $k_{\text{inj}}$ [-] | 0.20, **0.45**, 0.60 | Fraction of damaged cells after injury, Eq. (2) | [16,53] |
| $k_{\text{aga}}$ [$10^{-21}$ mol] | 0.125, **0.250**, 0.375 | Aggrecanase release from damaged cells, Eqs. (5) and (S15) | [62], model fit |
| $k_{\text{cl}}$ [$10^{-6}$ s$^{-1}$] | 0.3, **1.5**, 7.5 | Rate of strain-driven cell damage during cyclic loading, Eqs. (4), (S13), and (S14) | model fit |
| $k_{\text{cl,fl}}$ [$10^{-6}$ s$^{-1}$] | 0.3, **1.5**, 7.5 | Rate of fluid flow-driven aggrecan depletion, Eq. (S18) | model fit |
| $\hat{p}_{\text{synth,init}}$ [MPa $\cdot$ s$^{-1}$] | 10, **20**, 50 | Threshold for accelerated aggrecan biosynthesis (change of fluid pressure over time), Eq. (S21) | [22,58,63–65], model fit |

depended on the magnitude of fluid flow velocity [7,47,94]. The addition of new aggrecan via biosynthesis $R_{\text{biosynthesis}}$ depended on the change of fluid pressure over time in areas with healthy cells [9] experiencing moderate fluid pressure alterations. Here, the rate of change of fluid pressure considered to be in the moderate/healthy range of 20–60 MPa $\cdot$ s$^{-1}$ (ramp-up time of the cyclic loading 0.2s, 1 Hz, 40% duty cycle) corresponds to intermittent peak pressures in the range of 4–12 MPa that have been linked to anabolic responses in experiments [22,58,63–65]. The maximum increase in the aggrecan biosynthesis rate was set to 100% [14,40,71]. The sink/source terms $R_{\text{fluid flow}}$ and $R_{\text{biosynthesis}}$ (Table 1) included the main loading-related variables (fluid velocity and change of fluid pressure over time, respectively) as normalized, threshold-based functions (same form as Eq. (1), see Section C in S1 Text).

Due to limited experimental evidence for the model parameter values, a sensitivity analysis was conducted to assess how the most relevant model parameters affected the estimates of aggrecan content. The reference parameters (Table 2) represent the INJ and INJ + CL models, which predicted the average aggrecan contents based on experimental data. The parameter value ranges were selected based on computational and experimental studies and/or by adjusting the reference value to obtain a reasonable fit to the upper/lower bounds of the confidence intervals. In addition, two different initial aggrecan concentrations [7,62] and five different lesion geometries were investigated. These lesion models were generated by (1) segmenting five lesion geometries from images of histological sections cut from five different injured cartilage plugs, then (2) subtracting the segmented lesion geometry from the superficial–middle region of an intact rectangle (see Section 4.5).

## Supporting information

**S1 Text. Electronic supplementary material containing more detailed information. Fig** A. **Mesh sensitivity analysis. Fig** B. **Comparison of logarithmic axial and minimum principal strains. Table** A. **Material parameters in the mechanical loading model. Table** B. **Model parameters in the mechanoinflammatory cartilage adaptation model.** (DOCX)

## Acknowledgments

Eija Rahunen (Institute of Biomedicine, Cell and Tissue Imaging Unit, UEF) is acknowledged for the processing of the samples designated for digital densitometry analysis.

## Author contributions

**Conceptualization:** Atte S. A. Eskelinen, Joonas P. Kosonen, Moustafa Hamada, Amir Esrafilian, Cristina Florea, Alan J. Grodzinsky, Petri Tanska, Rami K. Korhonen.

**Data curation:** Atte S. A. Eskelinen.

**Formal analysis:** Atte S. A. Eskelinen.

**Funding acquisition:** Alan J. Grodzinsky, Rami K. Korhonen.

**Investigation:** Atte S. A. Eskelinen.

**Methodology:** Atte S. A. Eskelinen, Joonas P. Kosonen, Moustafa Hamada, Amir Esrafilian, Cristina Florea, Alan J. Grodzinsky, Petri Tanska, Rami K. Korhonen.

**Project administration:** Alan J. Grodzinsky, Rami K. Korhonen.

**Resources:** Alan J. Grodzinsky, Rami K. Korhonen.

**Supervision:** Cristina Florea, Alan J. Grodzinsky, Petri Tanska, Rami K. Korhonen.

**Visualization:** Joonas P. Kosonen, Cristina Florea, Petri Tanska, Rami K. Korhonen.

**Writing – original draft:** Atte S. A. Eskelinen.

**Writing – review & editing:** Atte S. A. Eskelinen, Joonas P. Kosonen, Moustafa Hamada, Amir Esrafilian, Cristina Florea, Alan J. Grodzinsky, Petri Tanska, Rami K. Korhonen.

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
