## [Decision Letter · Decision Letter 0]

5 Mar 2025

PCOMPBIOL-D-24-01918

Time-dependent computational model of post-traumatic osteoarthritis to estimate how mechanoinflammatory mechanisms impact cartilage aggrecan content

PLOS Computational Biology

Dear Dr. Eskelinen,

Thank you for submitting your manuscript to PLOS Computational Biology. After careful consideration, we feel that it has merit but does not fully meet PLOS Computational Biology's publication criteria as it currently stands. Therefore, we invite you to submit a revised version of the manuscript that addresses the points raised during the review process.

Please submit your revised manuscript within 60 days May 05 2025 11:59PM. If you will need more time than this to complete your revisions, please reply to this message or contact the journal office at ploscompbiol@plos.org. Please include the following items when submitting your revised manuscript:

We look forward to receiving your revised manuscript.

Kind regards,

David M Pierce

Guest Editor

PLOS Computational Biology

Jason Haugh

Section Editor

PLOS Computational Biology

**Journal Requirements:**

At this stage, the following Authors/Authors require contributions: Atte S. A. Eskelinen, Joonas P. Kosonen, Moustafa Hamada, Amir Esrafilian, Cristina Florea, Alan J. Grodzinsky, Petri Tanska, and Rami K. Korhonen. Please ensure that the full contributions of each author are acknowledged in the "Add/Edit/Remove Authors" section of our submission form.

Potential Copyright Issues:

i) Please confirm (a) that you are the photographer of 1A, or (b) provide written permission from the photographer to publish the photo(s) under our CC BY 4.0 license.

ii) Figure 1A. Please confirm whether you drew the images / clip-art within the figure panels by hand. If you did not draw the images, please provide (a) a link to the source of the images or icons and their license / terms of use; or (b) written permission from the copyright holder to publish the images or icons under our CC BY 4.0 license. Alternatively, you may replace the images with open source alternatives. See these open source resources you may use to replace images / clip-art:

5) In the online submission form, you indicated that your data will be submitted to a repository upon acceptance. We strongly recommend all authors deposit their data before acceptance, as the process can be lengthy and hold up publication timelines. Please note that, though access restrictions are acceptable now, your entire minimal dataset will need to be made freely accessible if your manuscript is accepted for publication. This policy applies to all data except where public deposition would breach compliance with the protocol approved by your research ethics board. If you are unable to adhere to our open data policy, please kindly revise your statement to explain your reasoning and we will seek the editor's input on an exemption.

1)  If the funders had no role in your study, please state: "The funders had no role in study design, data collection and analysis, decision to publish, or preparation of the manuscript."

7) Please ensure that the funders and grant numbers match between the Financial Disclosure field and the Funding Information tab in your submission form. Note that the funders must be provided in the same order in both places as well. Currently, " Doctoral Programme in Science, Technology and Computing (LUMETO) of the University of Eastern Finland" and "Strategic Funding of the University of Eastern Finland" are missing from the Funding Information tab. In addition, the order of the funders is not the same in both places.

**Reviewers' comments:**

Reviewer's Responses to Questions

Reviewer #1: In this manuscript, the authors develop a coupled mechanical and biological model that predicts aggrecan depletion in explants with and without an impact-induced focal lesion under cyclic loading. The mechanical model is well-established, using the porohyperviscoelastic framework and material constants previously employed by the research group for explants and whole joints. The biological model includes cell damage that induces aggrecanase release (causing proteolytic decreases in aggrecan content), fluid-flow driven aggrecan release, and pressure-driven aggrecan synthesis. Model parameters are obtained from a combination of prior literature and model fits; a sensitivity study was used to determine that the initial aggrecan content and lesion geometry are the most critical parameters to have accurate values for. The overall model results recapitulate experimental results, showing greater aggrecan depletion near lesions.

Overall, the manuscript and the approach have many strengths. The approach, while having parameters that are not well-established in the literature, is thoroughly documented with references and support provided to the extent available. The model is rigorously developed, including mesh sensitivity studies, parameter studies, and evaluation of sensitivity to geometry.

Despite these strengths, there are a few overarching concerns that should be more completely addressed in the manuscript. First, the aggrecan focus is narrow; the authors have prior work that also focuses on collagen failure. Why is collagen failure not included in this model? Second, the focus on immature cartilage samples hampers translational ability. While I appreciate that mature tissue can be more challenging to obtain (and more heterogeneous to work with), further discussion on what the steps are to translate this work into predicting aggrecan depletion in human cases would be useful (i.e., line 248-249 is insufficient). Third, the lack of experimentally-based constants is a real limitation of the literature; and the fact that some of the references are prior modeling studies (not experimental studies) should be made clear in the text.

Additional specific comments regarding the manuscript are addressed below.

-----Introduction

1 – The second paragraph is particularly well-written. The introduction overall is excellent.

2 – Line 102 “are presently few but” appears to be a typo.

-----Results

3 – Section 2.2: a pointer to the table with constants would be helpful here, as thresholds are mentioned but not given (e.g., “excessive shear strain: in line 136).

4 – Some of the threshold references do not seem applicable. For example, reference 61 does not provide a fluid pressure that would cause a response, it instead evaluates when behavior is incompressible.

-----Discussion

5 – Lines 258-262: why is the strain damage threshold larger in the model (40%) than in prior studies (7-12%)? Is this an artifact of other errors in the model?

6 – Lines 274-275: why use axial strain (which depends on coordinate system) rather than a principal strain (which is not dependent on coordinate system)?

-----Methods

7 – Line 366: is there ever lift-off in the bioreactor?

8 – Section 4.5: a recent publication described the potential errors in FE-predicted stress and strain in cartilage explants as a result of imperfections in the samples (https://doi.org/10.1016/j.jbiomech.2024.112323). Were your models idealized, and how does that compare to their actual geometry?

9 – Section 4.5: how was the lesion modeled? (And, in section 4.6, how were the five lesion models generated?)

-----Figures

10 – Figure 1: how far away from the center of the lesion is “away from lesions”?

Reviewer #2: This was an interesting, comprehensive manuscript. My main concern relates to the mechanoinflammatory model:

1. Mechanoinflammatory model. This is a novel aspect of the work and I found section 4.6 lacking in detail for me to appreciate this modelling component. I think the supplementary material S3 would be better placed in the main body of the manuscript. [S1 &S2 fine in the supplementary material]

2. The results section included temporal evolution of the aggrecan content, however didn't include temporal evolution of other quantities in the mechanoinflammatory (MI) model. It would be helpful to present temporal evolution of all variables in the MI (perhaps as supplementary material if it doesn't seem appropriate to put in main body of text)

3. A minor point on the MI model - there is no consideration of basal production/loss terms for the healthy/damaged cell populations. Over the time-scales of application of the model this doesn't seem relevant, however some comment here would be helpful why such terms are omitted - and if those terms may be relevant for modelling OA progression or rehabilitation modelling over longer timescales.

Reviewer #3: The authors developed a simplified mechanobiological model that combines mechanical loading effects with cellular responses in cartilage. The four elements of the model are cell "damage", proteolytic activity, fluid flow, and biosynthetic responses. The model predicted that fluid flow drives aggrecan loss near lesions, an effect outweighing that of proteolytic activity. The model also predicted biosynthetic responses outweigh degradation in deeper tissue.

Parts of the model involve speculation, mostly well acknowledged by the authors. The reviewer would have made different choices for several parts of the model. However, the reviewer is enthusiastic about this as a founding contribution in an important area and has only a few minor, optional suggestions for the packaging of the final manuscript.

Would the authors kindly consider adding discussion about the following limitations and adding a few notes about why specific choices were made?

1. The model only includes two cell populations ("healthy" and "damaged"), which misses mechanisms like apoptosis, necrosis, oxidative stress, or hypertrophy. How might more realistic cell models be incorporated?

2. The model omits tissue responses to injury-related pro-inflammatory cytokines and their effects on mechanical properties, which are known to affect osteoarthritis progression. What challenges led the authors to ignore these?

3. The model presumes an initial injury. What additional challenges might be faced with adding injury initiation?

4. Timescales are of course totally arbitrary when you don't have a great lock on rate constants. Why did the authors choose to simulate over only 12 days? This reviewer is no cow, but his own osteoarthritis has developed over years. How did the authors come up with these timescales?

In summary, this is a very interesting paper and the reviewer is highly enthusiastic.

**Have the authors made all data and (if applicable) computational code underlying the findings in their manuscript fully available?**

Reviewer #1: Yes

Reviewer #2: Yes

Reviewer #3: Yes

PLOS authors have the option to publish the peer review history of their article (what does this mean?). If published, this will include your full peer review and any attached files.

Reviewer #1: No

Reviewer #2: No

Reviewer #3: No

**Figure resubmission:**
---

## [Decision Letter · Decision Letter 1]

20 Oct 2025

Dear Mr. Eskelinen,

We are pleased to inform you that your manuscript 'Time-dependent computational model of post-traumatic osteoarthritis to estimate how mechanoinflammatory mechanisms impact cartilage aggrecan content' has been provisionally accepted for publication in PLOS Computational Biology.

Best regards,

Feilim Mac Gabhann, Ph.D.

Editor-in-Chief

PLOS Computational Biology

David Pierce

%CORR_ED_EDITOR_ROLE%

PLOS Computational Biology

Reviewer's Responses to Questions

**Comments to the Authors:**

Reviewer #1: The authors have addressed all comments.

Reviewer #2: Authors have addressed all concerns.

**Have the authors made all data and (if applicable) computational code underlying the findings in their manuscript fully available?**

Reviewer #1: None

Reviewer #2: Yes

PLOS authors have the option to publish the peer review history of their article (what does this mean?). If published, this will include your full peer review and any attached files.

Reviewer #1: No

Reviewer #2: No

---

## [Editor Report · Acceptance letter]

PCOMPBIOL-D-24-01918R1

Time-dependent computational model of post-traumatic osteoarthritis to estimate how mechanoinflammatory mechanisms impact cartilage aggrecan content

Dear Dr Eskelinen,

I am pleased to inform you that your manuscript has been formally accepted for publication in PLOS Computational Biology. Your manuscript is now with our production department and you will be notified of the publication date in due course.

With kind regards,

Anita Estes
